# Defect of *TIMP4* Is Associated with High Myopia and Participates in Rat Ocular Development in a Dose-Dependent Manner

**DOI:** 10.3390/ijms242316928

**Published:** 2023-11-29

**Authors:** Wenhui Zhou, Zixuan Jiang, Zhen Yi, Jiamin Ouyang, Xueqing Li, Qingjiong Zhang, Panfeng Wang

**Affiliations:** State Key Laboratory of Ophthalmology, Zhongshan Ophthalmic Center, Sun Yat-Sen University, Guangdong Provincial Key Laboratory of Ophthalmology and Visual Science, Guangzhou 510000, China; zhouwh23@mail2.sysu.edu.cn (W.Z.); jiangzx23@mail2.sysu.edu.cn (Z.J.); yizhen2015@sina.com (Z.Y.); ouyangjiamin@gzzoc.com (J.O.); drleexq@126.com (X.L.)

**Keywords:** *TIMP4*, form deprivation, myopia, rat model, genetic, extracellular matrix, bipolar cell

## Abstract

Thinning of the sclera happens in myopia eyes owing to extracellular matrix (ECM) remodeling, but the initiators of the ECM remodeling in myopia are mainly unknown. The matrix metalloproteinase (MMPs) and tissue inhibitors of matrix metalloproteinase (TIMPs) regulate the homeostasis of the ECM. However, genetic studies of the MMPs and TIMPs in the occurrence of myopia are poor and limited. This study systematically investigated the association between twenty-nine genes of the TIMPs and MMPs families and early-onset high myopia (eoHM) based on whole exome sequencing data. Two *TIMP4* heterozygous loss-of-function (LoF) variants, c.528C>A in six patients and c.234_235insAA in one patient, were statistically enriched in 928 eoHM probands compared to that in 5469 non-high myopia control (*p* = 3.7 × 10^−5^) and that in the general population (*p* = 2.78 × 10^−9^). Consequently, the *Timp4* gene editing rat was further evaluated to explore the possible role of *Timp4* on ocular and myopia development. A series of ocular morphology abnormalities in a dose-dependent manner (*Timp4*^−/−^ < *Timp4*^+/−^ < *Timp4*^+/+^) were observed in a rat model, including the decline in the retinal thickness, the elongation in the axial length, more vulnerable to the form deprivation model, morphology changes in sclera collagen bundles, and the decrease in collagen contents of the sclera and retina. Electroretinogram revealed that the b-wave amplitudes of *Timp4* defect rats were significantly reduced, consistent with the shorter length of the bipolar axons detected by HE and IF staining. Heterozygous LoF variants in the *TIMP4* are associated with early onset high myopia, and the *Timp4* defect disturbs ocular development by influencing the morphology and function of the ocular tissue.

## 1. Introduction

Myopia is an epidemic worldwide, with an increasing incidence rate in recent years [1]. High myopia (HM) is a severe form of myopia defined as a refraction error greater than −6.00 diopters (D) or an axial length greater than 26 mm. HM is often accompanied by pathological changes such as scleral thinning, retinal choroidal atrophy, posterior scleral staphyloma, macular degeneration, and choroidal neovascularization, and has become one of the leading causes of blindness [2]. While genetic and environmental factors play essential roles in the occurrence and development of myopia, early-onset high myopia (eoHM) is mainly caused by genetic factors occurring before school age [3,4]. To date, 27 loci (MYP1-MYP28, MYP4 was deprived) and ten genes (*OPN1LW,* OMIM 300822; *SCO2*, OMIM 604272; *ZNF644*, OMIM 614159; *CCDC111*, OMIM 615421; *LRPAP1*, OMIM 104225; *SLC39A5*, OMIM 608730; *P4HA2*, OMIM 600608; *ARR3*, OMIM 301770, *CPSF1*, OMIM 606027; *LOXL3*, OMIM 607163) are identified to be associated with myopia by linkage analysis combined with whole-exome sequencing (WES) or WES alone, which account for less than 15% patients with eoHM [5,6]. Additionally, variants in RetNet research have been identified in approximately 25% of the eoHM probands based on our previous studies [7,8]. The genetic factors for about 60% of the cause of eoHM are still unknown. Genome-wide association studies on high myopia have reported a package of related single-nucleotide polymorphisms (SNPs), but the role of these SNPs or genes harboring SNPs in the mechanism of myopia is largely unknown [9].

The occurrence and development of myopia are strongly related to scleral matrix remodeling [10] and excessive axial length elongation [11]. The scleral changes are accompanied by reduced synthesis and excessive scleral extracellular matrix (ECM) degradation. Collagen is a primary component of the ECM, and collagen disease is often accompanied by pathological high myopia such as Stickler syndrome and Marfan syndrome [12,13] because of its essential role in maintaining the structural integrity of ocular tissues such as cornea, retina, and sclera [14]. The homeostasis of the ECM is mainly regulated by an enzyme balance system, including the matrix metalloproteinase (MMPs), which degrade the ECM proteins, and the tissue inhibitor of matrix metalloproteinase (TIMPs), which inhibit the MMPs [15]. Twenty-nine genes (including the MMP 23A and MMP 23B) of the MMPs and four TIMPs have been found in vertebrates [16]. Several investigations reported an increase of the TIMPs and MMPs in aqueous humor in high myopia patients [17], where the TIMPs served as a compensatory response to inhibit the MMP-induced degradation of the ECM, thereby constraining myopia’s progression [18]. MMP-2 significantly upregulated in myopia animal models such as tree shrews, chickens, and mice [19,20,21]. The dysregulation of MMP/TIMP balance may contribute to myopia development, but none have been genetically studied in myopia patients.

This study aims to explore the relationship between the thirty-three MMPs/TIMPs genes and eoHM. We further probed the underlying mechanisms by gene mutant rat model.

## 2. Results

### 2.1. The TIMP4 Loss-of-Function Variants Clustered in Individuals with High Myopia

We screened variants in twenty-nine TIMP and MMP genes (Table A1) using WES data from 928 eoHM individuals without related syndromic diseases and 5469 non-HM individuals with different eye conditions. Two heterozygous loss-of-function (LoF) variants in the *TIMP4* gene were detected in probands from seven unrelated families (Table 1), including c.528C>A (p.C176*) in six probands and c.234_235insAA (p.M80Rfs*26) in one proband (Figure 1A). None of the other known causative variants for HM or genetic eye disease were detected in these probands. The frequency of the two heterozygous LoF variants in the *TIMP4* was 3.77 × 10^−3^ (7/1856) in our 928 eoHM cohort, 1.83 × 10^−4^ (2/10938) in the 5469 non-HM participants, 9.78 × 10^−5^ (26/285849) in gnomAD overall population (Figure 1B and Figure A2B) and was 7.06 × 10^−4^ (14/19843) in the gnomAD East Asia population. The frequency of the LoF *TIMP4* allele in the HM cohort was significantly higher than that in non-HM populations (Fisher’s exact test, *p* = 3.7 × 10^−5^) and in the gnomAD population (Fisher’s exact test, *p* = 2.78 × 10^−9^) as well as in the East Asia population (Fisher’s exact test, *p* = 1.0 × 10^−3^) (Figure 1B). The LoF variants in the *TIMP4* were relevant to eoHM with the mean OR value of 20.70 (CI: 4.30–99.73, *p* = 3.70 × 10^−5^), indicating that individuals with the *TIMP4* LoF variants are approximately 20.7 times more likely to develop eoHM compared to those without the LoF variants. Ocular phenotypes of 7 probands with the heterozygous LoF variants were shown in Table 1. The spherical equivalent (SE) fraction was measured to be over −6.00 D in all probands. The typical pathological changes in high myopia were observed, including tigroid fundus, temporal crescent of the optic nerve, and latency delay in the visual evoked potential (VEP) (Table 1, Figure 1C). Co-segregation was observed in family P1, in which the father of proband 1 (P1) carrying heterozygous c.528C>A began to develop myopia at the age of seven and exhibited a SE of −18.00 D in both eyes, along with an ocular axial length of over 34 mm. The mother of P1, who did not harbor this variant, was not diagnosed with high myopia.

The nucleotide change at site 528 encodes amino acid No. 176, which participates in disulfide bond formation and plays a crucial role in protein stability. The frameshift variant c.234_235insAA was predicted to cause premature termination of protein coding, triggering nonsense-mediated mRNA decay (NMD), according to several in silico bioinformatic tools. The two variants are both in the Netrin Module (NTR) domain (Figure 1D) and are highly conserved across vertebrates (Figure A1A). The bioinformatics tool predicted that both variants would dramatically change the corresponding three-dimensional peptide structure of the TIMP4 and might result in an incomplete main structure of the protein, including the NTR domain, and thus pathogenic (Figure 1D).

### 2.2. Timp4 Gene Editing Rat

Gene-editing rats were constructed to evaluate the potential functional roles of the *TIMP4* on eyeball development. The exon 1 and exon 5 of the *Timp4* of rats were selected as target sites of CRISPR/Cas9-mediated genome engineering, which resulted in the loss of the whole *Timp4* gene in the genome (Figure 2A). Sanger sequencing confirmed the absence of fragment flanking exon 1 to exon 5 of the *Timp4* based on genomic DNA extracted from the blood from the tail of the *Timp4*-gene editing rat (Figure 2B). Three kinds of genotypes, *Timp4^+/+^*, *Timp4^+/−^*, and *Timp4^−/−^*, are present by DNA electrophoresis (Figure 2C). Western blot revealed that the expression of the *Timp4* in three-week-old rats’ retinas dramatically decreased after knock-out compared with wild-type (Figure A2A).

### 2.3. Timp4 Expression in the Retina

To identify the tissue and cell types in which the TIMP-4 protein is located, we further investigated its distribution in the human and rat retina at the protein level by immunohistochemistry (IF). In the human retina, immunostaining was predominantly observed in the outer plexiform layer (OPL) and inner nuclear layer (INL) (Figure 2D). In the rat retina, there was a higher fluorescence density at the OPL with weak staining in INL (Figure 2D). TIMP4 is mainly found in the OPL, which consists of a dense network of synapses from bipolar cells and photoreceptor cells.

### 2.4. Ocular Morphology and Electroretinography Changes in Timp4^−/−^ Rats in Normal Development

The CT, ACD, VCD, RT, VCD, and axial length (AL) of *Timp4*^+/+^, *Timp4*^+/−^, and *Timp4*^−/−^ rats were measured throughout 3 to 52 weeks (Table A2). The RT of *Timp4*^−/−^ rats was thinner than those of *Timp4*^+/+^ rats and decreased in a *Timp4* dose-dependent manner (*Timp4*^−/−^ < *Timp4*^+/−^ < *Timp4*^+/+^). These significant differences of RT were observed at four timepoints including 3W (*Timp4*^+/+^ = 0.2382 ± 0.0121 mm, *Timp4*^+/−^ = 0.2290 ± 0.0054 mm, *Timp4*^−/−^ = 0.2212 ± 0.0075 mm, *p* = 3.6 × 10^−5^), 6W (*Timp4*^+/+^ = 0.2233 ± 0.0147 mm, *Timp4*^+/−^ = 0.2171 ± 0.0086 mm, *Timp4*^−/−^ = 0.2073 ± 0.0055 mm, *p* = 4.4 × 10^−4^ < 0.001), 9W (*Timp4*^+/+^ = 0.2148 ± 0.0051 mm, *Timp4*^+/−^ = 0.2052 ± 0.0062 mm, *Timp4*^−/−^ = 0.2012 ± 0.0088 mm, *p* = 0.013 < 0.05), 10W (*Timp4*^+/+^ = 0.2103 ± 0.0061 mm, *Timp4*^+/−^ = 0.2051 ± 0.0066 mm, *Timp4*^−/−^ = 0.2009 ± 0.0083 mm, *p* = 0.022 < 0.05). The thinner retina was presented in a *Timp4* dose-dependent manner, similar to other myopia gene-editing animal models [22]. The axial length of *Timp4*^−/−^ rat was longer than those of *Timp4*^+/+^ and *Timp4*^+/−^ rats in 3 and 9 weeks (3W: *Timp4*^+/+^ = 5.143 ± 0.025 mm, *Timp4*^+/−^ = 4.984 ± 0.080 mm, *Timp4*^−/−^ = 5.246 ± 0.033 mm, *p* = 1.04 × 10^−5^ < 0.001; 9W: *Timp4*^+/+^ = 5.923 ± 0.084 mm, *Timp4*^+/−^ = 5.891 ± 0.119 mm, *Timp4*^−/−^ = 5.892 ± 0.094 mm, *p* = 0.007 < 0.01) (Figure 2E). Also, a visual tendency of VCD was observed in the different groups from 3 weeks to 52 weeks (*Timp4*^−/−^ > *Timp4*^+/−^ > *Timp4*^+/+^), but no significant difference was found yet. No significant differences or tendencies were found among the three groups at most time points of the other ocular parameters, including CT, ACD, and LT (Figure A3). 

ERG changes in the three groups were assessed at three weeks and six weeks of age, corresponding to the time points of the remarkable eye tissue parameter differences observed in the Timp4 gene editing rats. Analysis of scotopic ERG (dark-adapted ERG) in the Timp4 gene editing rats revealed that the b-wave amplitude was reduced in both the Timp4-gene editing rats compared to wild-type rats at 3 and 6 weeks of age under different intensities. A significant difference was observed at six weeks of age (One-way ANOVA, *Timp4*^−/−^ < *Timp4*^+/−^ < *Timp4*^+/+^, 0.01 log scotopic cd.s/m^2^: *Timp4*^+/+^ = 150.17 ± 31.32 μV, *Timp4*^+/−^ = 106.31 ± 29.94 μV, *Timp4*^−/−^ = 118.70 ± 31.21 μV, *p* = 0.037 < 0.05; 3.0 log scotopic cd.s/m^2^: *Timp4*^+/+^ = 234.33 ± 37.58 μV, *Timp4*^+/−^ = 193.30 ± 21.72 μV, *Timp4*^−/−^ = 165.63 ± 6.41 μV, *p* = 0.007 < 0.01; 10.0 log scotopic cd.s/m^2^: *Timp4*^+/+^ = 202.67 ± 29.14 μV, *Timp4*^+/−^ = 175.60 ± 20.67 μV, *Timp4*^−/−^ = 155.11 ± 9.77 μV, *p* = 0.029 < 0.05). No significant difference was observed in the a-wave amplitude among the groups (Figure 2F). 

### 2.5. Ocular Morphology Changes of Timp4 Defect Rats under form Deprivation Pressure

Since the axial length elongation was observed in the *Timp4* gene editing rats discontinuously, we burdened the environment pressure of form deprivation (FD) model on gene-editing rats to enlarge the possible influence of the *Timp4* knockdown on ocular development (Figure A2C). Fifteen rats from three genotypes were used (n = 5, respectively). After wearing a translucent diffuser for 72 h (Figure 3A), ocular change was recorded by comparing the value before FD and after FD in both experimental eyes (OD) and the control eyes (OS) as previously reported [23] (ocular change = value after FD minus value before FD). Changes of different ocular parts, including ACD, VCD, RT, and AL, were not significant in wild-type rats but were significant in *Timp4*-deficient rats (Paired T-test, change in ACD: *Timp4*^+/+^, *p* = 0.412; *Timp4*^+/−^, *p* = 0.644; *Timp4*^−/−^, *p* = 0.007 < 0.01; change in VCD: *Timp4*^+/+^, *p* = 0.310; *Timp4*^+/−^, *p* = 0.654; *Timp4*^−/−^, *p* = 0.008 < 0.01; change in RT: *Timp4*^+/+^, *p* = 0.901; *Timp4*^+/−^, *p* = 0.194; *Timp4*^−/−^, *p* = 0.027 < 0.05; change in AL: *Timp4*^+/+^, *p* = 0.680; *Timp4*^+/−^, *p* = 5.29 × 10^−5^ < 0.001; *Timp4*^−/−^, *p* = 0.521) (Figure 3B). After FD modeling, FD eyes of gene defect rats showed a significant reduction of bipolar axon length in the central and mid-peripheral retina compared to non-FD eyes, while similar differences were not observed in wild-type rats (Paired T-test, central: *Timp4*^+/+^ *p* = 0.1129, *Timp4*^+/−^ *p* = 0.0398 < 0.05, *Timp4*^−/−^ *p* = 0.392; mid-peripheral: *Timp4*^+/+^ *p* = 0.1674, *Timp4*^+/−^ *p* = 0.0087 < 0.01, *Timp4*^−/−^ *p* = 0.0071 < 0.01; peripheral: *Timp4*^+/+^ *p* = 0.1065, *Timp4*^+/−^ *p* = 0.6741, *Timp4*^−/−^ *p* = 0.1219) (Figure 3C,D). Detailed images are provided in Figure A4.

### 2.6. Content and Morphology Change in the Collagen of Timp4 Defect Rats

Since collagen is involved in myopia progression and is the target factor of *TIMP4*, we further evaluated the ocular tissues’ collagen to explore whether *TIMP4* is related to a reduction in collagen content. In the *Timp4*-knockdown rats, the collagen contents of the sclera, cornea, and retina were significantly lower than those in wild-type rats (sclera: *Timp4*^+/+^ = 2719.9 ± 535.7 μg/mL, *Timp4*^+/−^ = 2112.9 ± 167.4 μg/mL, *Timp4*^−/−^ = 1941.2 ± 190.0 μg/mL, *p* = 0.047 < 0.05; cornea: *Timp4*^+/+^ = 854.2 ± 156.4 μg/mL, *Timp4*^+/−^ = 676.3 ± 148.5 μg/mL, *Timp4*^−/−^ = 522.4 ± 101.4 μg/mL, *p* = 0.031 < 0.05; retina: *Timp4*^+/+^ = 62.8 ± 21.5 μg/mL, *Timp4*^+/−^ = 33.5 ± 5.5 μg/mL, *Timp4*^−/−^ = 32.6 ± 5.8 μg/mL, *p* = 0.025 < 0.05) (Figure 4A, Table A3). No significant differences were found in the lens and vitreous body. The ultrastructural morphology of collagen fiber of the sclera area surrounding the optic nerve was observed by using electron microscopy, and the results showed that in wild-type rats, the collagen fiber bundles had smooth edges, relatively uniform thickness, and an average diameter of approximately 114.5 μm, with smaller gaps and a tight arrangement between adjacent fibers. The sclera of *Timp4*-deficient rats exhibited collagen fibers with diameters of about 82.8 μm, varying sizes, unsmooth edges, and more significant gaps between adjacent collagen fibers. The statistical analysis revealed that the sclera collagen fiber diameter was significantly reduced in the *Timp4*-gene editing rats compared to wild-type rats (*Timp4*^+/+^ = 114.5 ± 29.22, *Timp4*^+/−^ = 103.4 ± 20.21, *Timp4*^−/−^ = 82.77 ± 26.68, *p* < 0.0001) (Figure 4B, Table A4). The alterations in collagen fiber structure in the *Timp4*-deficient rats may have implications for the biomechanical properties of the sclera, such as its tensile strength, elasticity, and resistance to deformation, which is closely related to myopia formation.

### 2.7. Morphology Changes in the Retina of Timp4^−/−^ Rats

The result of the HE staining revealed that the thickness of the retina decreased with increasing age across different genotypes (Figure 5A). The thickness of the central retina of *Timp4*^+/−^ and *Timp4*^−/−^ rats was significantly decreased compared to that of the *Timp4*^+/+^ rats from 3 w to 9 w (*Timp4*^−/−^ < *Timp4*^+/−^ < *Timp4*^+/+^) (Figure 5B, Table A5), consistent with the SD-OCT results. At 3W and 6W, the central retina thickness is significantly thinner in the *Timp4*-deficient rats than that in wild-type rats (3W central retina thickness: *Timp4*^+/+^ = 208.22 ± 3.76 μm, *Timp4*^+/−^ = 187.44 ± 7.97 μm, *Timp4*^−/−^ = 175.87 ± 11.31 μm, *p* = 3.41 × 10^−6^; 6W central retina thickness: *Timp4*^+/+^ = 166.67 ± 12.29 μm, *Timp4*^+/−^ = 153.95 ± 13.73 μm, *Timp4*^−/−^ = 137.65 ± 13.53 μm, *p* = 0.006 < 0.01). At 9W, the central and mid-peripheral retina is thinner in *Timp4*-deficient rats than that in the wild-type (9W central retina thickness: *Timp4*^+/+^ = 149.47 ± 8.8 μm, *Timp4*^+/−^ = 133.23 ± 2.73 μm, *Timp4*^−/−^ = 129.07 ± 6.1 μm, *p* = 4.27 × 10^−4^; 9W mid-peripheral retina thickness: *Timp4*^+/+^ = 122.23 ± 2.4 μm, *Timp4*^+/−^ = 138.03 ± 4.91 μm, *Timp4*^−/−^ = 114.13±5.75 μm, *p* = 4.22 × 10^−5^). The differences in retinal thickness between different groups were mainly observed in the bipolar cells and photoreceptor cell layer of the retina, specifically in the IPL to INL (Table A6). Thus, we further observed the changes in ON bipolar cells whose axon is in IPL-INL. IF analysis showed that PKC-α area was shorter in the *Timp4* defect rats in the central and mid-peripheral retina (central: *Timp4*^+/+^ = 64.91 ± 4.58 μm, *Timp4*^+/−^ = 54.06 ± 8.62 μm, *Timp4*^−/−^ = 49.06 ± 2.26 μm, *p* = 0.0155 < 0.05; mid-peripheral: *Timp4*^+/+^ = 76.01 ± 5.62 μm, *Timp4*^+/−^ = 60.25 ± 0.91 μm, *Timp4*^−/−^ = 56.90 ± 11.48 μm, *p* = 0.041 < 0.05, peripheral: *Timp4*^+/+^ = 50.34 ± 3.81 μm, *Timp4*^+/−^ = 50.41 ± 1.12 μm, *Timp4*^−/−^ = 49.09 ± 7.91 μm, *p* = 0.94), which indicated that the gene defect rats had shorter bipolar cells axons in central and mid-peripheral regions of the retina compared to wild-type rats (Figure 6). The decrease of b-wave revealed by ERG was further corroborated by structural changes observed in the IF and HE staining.

## 3. Discussion

This study identified two *TIMP4* LoF variants of the *TIMP4* in seven of 928 probands with early-onset high myopia, and the defect of the *TIMP4* was significantly enriched in eoHM compared to 5469 non-myopia controls (*p* = 3.7 × 10^−5^) and the general population (*p* = 2.78 × 10^−9^). We further confirmed the possible role of the *TIMP4* by gene defect rat model and found that several changes in ocular structure and function were observed in the *Timp4* defect rats in a dose-dependent manner, including the thinner retina, reduced b-wave amplitude, shortened axon length of bipolar cells, smaller sclera collagen fibers, and decreased collagen content in the sclera, retina, and cornea. Extra pressure induced by FD modeling would amplify these structure and function changes. These findings suggested that a defect of the *TIMP4* is associated with high myopia and participates in rat ocular development in a dose-dependent manner.

The TIMP-4 belongs to the TIMP family, a group of structurally similar proteins that inhibit the activity of metalloproteinases. Unlike other family members (TIMP1, TIMP2, TIMP3), the role of TIMP-4 in ocular tissue remains limited [24]. Previous studies revealed that the protein structure of the TIMP-4 can be inferred from the TIMP-2, sharing 51% homology at the amino acid level, including the location of 12 cysteine residues and a leader sequence comprising 29 amino acids, which may be the target sequence cleaved to generate the mature protein [16,25]. The TIMP-4 efficiently inhibits the activity of MMP-2 via its N-terminus [25,26], and MMP2 upregulation was accompanied by significant myopia [18]. Both the *TIMP4* LoF variants detected in our study are in the NTR domain, which is implicated in many protein functions such as axon guidance, regulation of the Wnt signaling pathway, and, notably, metalloproteinase activity [27,28]. Our findings linked these variants to high myopia based on the following reasons: (1) the *TIMP4* LoF variants are enriched in the high myopia cohort in our study; (2) the TIMP4 LoF variants were predicted to result in truncation changes, disrupting domain with important biological functions; (3) pathological changes resembling those in HM individuals were observed in the *Timp4*-knockdown rat model in a dose-dependent manner; (4) form deprivation could exacerbate the pathological changes in gene defect rats. Previous studies of extracellular matrix-related genes (the MMP and TIMP family) were likely driven by the known biological roles in tissue remodeling, possibly overlooking their molecular genetic analysis on patients with high myopia. While the TIMPs has been studied through WES in other contexts, such as schizophrenia [29], no systematical studies of ECM-associated high myopia were performed based on WES data before. This might explain why the LoF variants of the *TIMP4* were not previously associated with high myopia in large cohorts. Currently, there is no entry of human disease caused by *TIMP4* mutation listed in the OMIM database. To our knowledge, this study is the first to discover the role of *TIMP4* in high myopia, providing fresh insights into the etiology of myopia.

During myopia development, axial length elongation is regulated by multiple mechanisms across the retina with a retina-to-sclera direction [30,31]. The ECM of the sclera was remodeled as the consequence of a growth-regulating cascade initialed from the retina. Although 14 high myopia genes were reported (Table A7), only *CTSH* and *BSG*, reappeared with longer axial length in gene deficiency animals [32] without further mechanism explorations. In this study, we observed that the quality and quantity of collagen in the sclera of *Timp4^−/−^* rats were decreased simultaneously, which might address the elongation of vitreous and axial length. Our finding supports the view that ECM remodeling contributes to myopia development by changing the biomechanical properties of the sclera. Besides remodeling the sclera, *Timp4* also participated in the abnormality of bipolar cells in the retina. TIMP4 protein expressed from OPL to IPL of both human and rat where synapses of the bipolar cell were involved and the primary defect in the retina of *TIMP4*^−/−^ rat were also focused on bipolar cell including reduced amplitude of the b-wave as well as shorter bipolar cells axons in eyeball development and under FD. Total axon length (from the axon hillock under the soma to the axon terminal tip) and IPL axon length (measured by the distance from the INL-IPL border to the axon terminal) [33] were often used to explore the functional impact of the neuron in literature [34,35]. Retinal bipolar cells act as the “projection neurons” in the visual circuit of vertebrates and convey essential visual information via synaptic from photoreceptor to ON bipolar cells mediating by the mGluR6 receptor [36]. In our study, we measured PKCα area to find that *Timp4* defect rats have shorter axon length and reduced b-wave amplitude compared to wild-type rats, which indicates both functional and structural damage of bipolar cells in *Timp4* defect rats. The abnormalities of bipolar cells or neural channels in which they revolved have qualitatively different effects on ocular growth and myopia development [37,38,39,40]. ECM plays an essential role in retinal development by regulating cell shape, proliferation, differentiation, migration, and morphogenesis [41,42] and has been used to facilitate the differentiation of human neural stem cells into neurons and to increase axon length [43]. In Figure 4B, we present the frequency distribution of collagen fiber diameters, where a notable error bar in *Timp4^−/−^* rats suggests heterogeneity and dispersed distribution of fiber sizes, which aligns with the visual trends depicted in Figure 4A. Our study suggests a model wherein *Timp4* deficiency may lead to a change of ECM in the retina, which changes the function and structure of bipolar cells, resulting in high myopia.

Although rats are not a classical model for myopia study, we chose rats for this research considering the following reasons: (1) gene-editing rats are easy to obtain; (2) rats have eyes twice the size of mice and are convenient for performing FD modeling and collect ocular parameters; (3) rat as an animal model for FDM was successfully established in previous literature [44,45]. In this study, the purpose of FD was to discover whether or not the ocular structure and function changes caused by the *Timp4* defect in rats would be influenced by myopia-related pressure. After FD modeling, retina thickness and more extended vitreous chamber depth were observed, which was not exhibited in untreated gene deficiency rats. The model exhibited phenotypic features such as thinner retina thickness, shorter bipolar cell length, deeper VCD, and longer AL compared to the control eyes. The significant differences between FD and control eyes were not observed in wild-type rats. *Timp4*-defect animal models have mainly been studied in the cardiovascular system (Table A8). Although *TIMP4* is known to participate in physiological and pathological processes such as cell proliferation, differentiation, apoptosis, and extracellular matrix degradation, no studies have reported ocular changes associated with this gene in animal models or human diseases.

Interestingly, researchers discovered that in *Timp4*-defect animal models, its loss moderately compromised cardiac functions with aging, and gene-defect animals exhibited a higher susceptibility to mortality in induced myocardial infarction [46]. This pattern was also observed in our rat FDM models, as evidenced by retinal thinning during development and changes after FD modeling. To some extent, such a “coincidence” reveals this gene’s interactive nature with external environmental stress.

The association of *TIMP4* deficiency and myopia was robustly supported by findings based on myopia patients and the gene editing rat model, but our study had several limitations. First, although the *TIMP4* LoF variant enriched the HM population in our study, it was also detected in non-high myopia individuals and GenomAD. It remains to be seen whether incomplete penetrance happened in the pathogenic mechanisms of TIMP4 on eoHM based on more individuals with *TIMP4* LoF. Secondly, statistically significant differences in retinal thickness and axial length were concentrated in the adolescent period of *Timp4* deficiency rats. It is unknown whether the differences are the causal link to the special role of *Timp4* in the eye development stage or to the limited animal number at different time points after 15 weeks in this study (Figure 2E). Thirdly, both functional and structural damage was observed in bipolar cells from *Timp4* deficiency rats. However, it is unknown which subtypes of ON and OFF bipolar cells are involved, and the mechanism of how ECM affects the function of bipolar cells remains unknown. Further studies are necessary to answer these questions.

## 4. Materials and Methods

### 4.1. Patients and Clinical Information

This study was approved by the institutional review board of the Zhongshan Ophthalmic Centre. Participants with various ocular conditions, including early-onset high myopia, were recruited from the Pediatric and Genetic Clinic, Zhongshan Ophthalmic Centre, Guangzhou, China. Clinical information was gathered according to the established standard operation protocols [47]. The individuals with eoHM were selected based on criteria of high myopia that occurs before school age and is considered to be caused mainly by genetic variations with minimal environmental involvement [32]. The control groups of subjects with other ocular disorders and healthy controls were recruited as previous studies [48]. Before collecting clinical data and peripheral blood samples, written informed consent was obtained from the participants, their guardians, or relatives. The informed consent was obtained adhering to the tenets of the Declaration of Helsinki. Human retina sections were obtained from Zhongshan Ophthalmic Center eye bank under the Medical Ethics Committee of Zhongshan Ophthalmic Center, Sun Yat-Sen University (Approval number: 2011KYNL012).

### 4.2. Candidate Gene Detection by WES and Multistep Bioanalysis

Potential pathogenic variants (PPVs) in TIMPs and MMPs were selected from an in-house whole exome sequencing (WES) dataset comprising 928 probands with high myopia and 5469 probands with other forms of genetic eye diseases as our previous process [49]. The WES procedures and multistep bioinformatic analysis of detected variants were detailed in our previous studies [47]. The frequency of PPV in the HM cohort was calculated and compared with that of the non-HM genetic eye disease cohort and the general population database (gnomAD v2.1.1) to confirm the candidate gene where LoF variants were rare and intolerant. As described in our previous study, all PPVs were confirmed by Sanger sequencing in probands and available family members [47]. The PPVs were described according to the nomenclature for sequence variations (Human Genome Variation Society nomenclature v20.05, http://varnomen.hgvs.org/, accessed on 1 May 2022) and confirmed by Mutalyzer [50] (https://www.mutalyzer.nl/, accessed on 10 March 2023). Pathological classifications were performed according to the guidelines of the American College of Medical Genetics and Genomics (ACMG) [51]. Prediction of variants was conducted using ENTPRISE-X (http://cssb2.biology.gatech.edu/entprise-x/, accessed on 1 May 2022) and SIFT (https://sift.bii.a-star.edu.sg/www/SIFT_indels2.html, accessed on 1 May 2022). In silico prediction of protein structure was conducted using AlphaFold (https://alphafold.ebi.ac.uk/, accessed on 1 May 2022), DUET (http://biosig.unimelb.edu.au/duet/stability, accessed on 1 May 2022) and MaestroWeb (https://pbwww.che.sbg.ac.at/maestro/web, accessed on 1 May 2022).

### 4.3. Knockdown of Timp4 Gene in Rats

The experiments were approved by the Animal Experimental Ethical Inspection to Ethics Committee of Zhongshan Ophthalmic Center, Sun Yat-Sen University (Approval number: 2020-030, 2020-036, 2020-172). All procedures were strictly carried out per the ARVO statement for the Use of Animals in Ophthalmic and Vision Research and the guidance of the IACUC (Institutional Animal Care and Use Committee). The *Timp4* gene (GenBank accession number: NM_001109393.1; Ensembl: ENSRNOG00000007955) knockdown Sprague-Dawley rat (SD rat) model was commercially constructed by Cyagen Bioscience Inc. (Guangzhou, China) using CRISPR/Cas-mediated genome engineering. The construction and validation details are shown in Figure 2A–C, Cas9 mRNA and gRNA generated by in vitro transcription were injected into fertilized eggs for knock-out rat productions. The sequences of gRNA were as follows: gRNA1: 5′-TAGGGCCAATCCCGCCCCCATGG-3′; gRNA2: 5′-CTTCCATAGCGCACATCGCAAGG-3′. The primer sequences of genotyping were as follows: Rat *Timp4*-F: 5′-CCCTGAGTAGCTTGTCTCATTCCCAG-3′, Rat *Timp4*-R: 5′-AGAAACCTAACGGAACTGAAGAGAAGTCT-3′, Sequencing primer: 5′-GGTCCGTCCTCCTCTGTCACTGT-3′, and the reaction method of polymerase chain reaction were listed in Table A9 and Table A10. The positive founders (F0) genotyped by PCR and DNA sequencing analysis were then bred to the next generation (F1). We used littermates as controls to ensure genetic and environmental consistency in our experimental design. Three genotypes were included in this study: wild-type (*Timp4*^+/+^), heterozygous (*Timp4*^+/−^), and homozygous (*Timp4*^−/−^). All rats were transported under the standard procedure of the Ophthalmic Animal Laboratory in special containments equipped with filtration systems to maintain specific pathogen-free conditions. Upon arrival at the laboratory, the rats were subjected to a seven-day acclimatization period before being delivered to the animal facility. All animals were housed in the Ophthalmic Animal Laboratory, Zhongshan Ophthalmic Center, Sun Yat-sen University. Controlled photoperiods (12 h of light and 12 h of darkness) and sufficient water and food were provided in a temperature-controlled room in a specific pathogen-free environment. To ensure consistency, three-week-old rats of the same sex (male) and the same body weight (50 ± 10 g) were utilized. Abnormally growing rats were excluded from the study (body weight below or over -2 standard deviations of the group, Table A11). Western blot was conducted as previously reported [49]. The primary antibodies used in the western blot were TIMP-4 (PA5-92959, 1:1000, Invitrogen, Waltham, MA, USA) and Beta-actin (AC004, 1:1000, Abclonal, Wuhan, China). Peroxidase-conjugated secondary antibody antibodies were anti-rabbit IgG (4413, 1:2000, Cell Signaling Technology, Danvers, MA, USA) and anti-mouse IgG antibodies (4408, 1:2000, Cell Signaling Technology, USA) diluted by 1:2000.

### 4.4. Determination of Ocular Parameters

Axial length (AL), corneal thickness (CT), anterior chamber depth (ACD), vitreous chamber distance (VCD), and retinal thickness (RT) from three weeks to 52 weeks of age were measured by spectral-domain optical coherence tomography (SD-OCT, Envisu R4310, Leica, Germany) as previously described [52,53] (Figure A2B). Rats were administered 0.5% tropicamide and 0.5% phenylephrine eye drops (Santen, Japan) to induce mydriasis. After an intraperitoneal injection of Zoletil 50 (Virbac, Carros, France) at 75 mg/kg dosage for anesthesia. Rats were immobilized on a platform for scanning. The configuration detail was as follows: a scanning circle with a radius of 2.0 cm (axial resolution: 2.6 μm, central wavelength: 840 nm, 24 frames × 5 B scans, each B scan averaged five times) was centered on the macula. Accurate shots were obtained when a cross flash (signal targeting optic disc) appeared on the screen to ensure that the longest eye axis was captured each time. The images obtained from the scans were analyzed using InVivoVue software 2.4 or DIVER 3.4, and data around the optic papilla at 360° was measured to calculate the average value of CT, ACD, VCD, and RT. The lens thickness (LT) measurement was limited to rats aged between three and 15 weeks, as the distance from the cornea to the lens of older rats would be beyond the maximum range of R4310 (5.2 mm). The AL was calculated by summing the values of CT, ACD, LT, VCD, and RT.

### 4.5. Determination of Ocular Components under Extra Pressure from Form Deprivation

Several genes were reported to be associated with HM, but classical myopia characteristics such as refractive error change and prolonged axial length were not observed in gene mutant animal models [22,54]. The form deprivation myopia (FDM) was a classical approach to induce a high myopia model and it was successfully established in different animal species, including monkeys, mice, chickens, tree shrews, guinea pigs, and rats [44,55,56,57,58,59,60]. To amplify the potential impact of *Timp4* knockdown on ocular development, we established a form deprivation model on the three-week-old rats of three genotypes, referring to the previously reported form deprivation models of chicken and rats [45,61] (Figure A2B). Male rats were randomly grouped and anesthetized as described above. After ocular parameters measurement, a form deprivation (FD) translucent diffuser made of soft polyvinyl chloride (outside diameter: 20 mm; inside diameter: 10 mm) was fitted to the right eye (OD) of each rat by matching Velcro rings. The left eyes (OS) were untreated and served as control eyes. The diffusers utilized in this study were fabricated from a soft polyvinyl chloride material, designed to induce a blurring effect without causing harm to the ocular or facial tissues. Following anesthesia, the diffusers were gently affixed on the surface of the orbital skin by super glue (Pattex, Henkel, Düsseldorf, Germany). These lenses diffused light, resulting in the retina’s inability to form clear images, a condition hypothesized to contribute to the development of myopia. The diffusers were cleaned twice daily, and the rats whose diffusers fell off during the FDM modeling were excluded from the experimental group. After 72 h of the FDM modeling, the diffusers were removed, and the ocular components of both eyes were measured immediately using the SD-OCT.

### 4.6. Determination of Retina Function by Electroretinography

The retina’s function was evaluated using electroretinography (ERG) in experimental animals, as previously described [62]. Animals were divided into different groups as described above and tested at 3 and 6 weeks. Before the experiment, the rats were acclimated to complete darkness for at least eight hours, with access to sufficient food, water, and exercise space. The anesthetized rats were measured with a Roland electrophysiology instrument (RETI-port/scan 21, Roland Consult, Brandenburg an der Havel, Germany) at different scotopic levels (0.1, 3.0, and 10.0 log scotopic cd.s/m^2^) as well as oscillatory potentials. Each group consists of three to five animals.

### 4.7. Hematoxylin-Eosin (HE) Staining and Immunofluorescence (IF)

We conducted the HE and IF staining to validate the results from the SD-OCT and ERG and investigate changes occurring beyond the detection range. Tissues were harvested after execution by cardiac perfusion, and the eyes were enucleated and fixed with a fixative solution (FAS, G1109, Servicebio, Wuhan, China) for 24 h before preparing paraffin slices. The IF and HE staining were performed on sections prepared from paraffin-embedded tissue. During the embedding process, we orient the optic nerve parallel to the plane of sectioning to ensure maximal representation of the ocular axis on the sections. Tissues were sectioned by 4 μm. Sections containing the optic nerve were selected for subsequent staining procedures. Standard cardiac perfusion, HE, and IF staining procedures were conducted as previously described [49,63]. Due to the decreased b-wave amplitude observed in the functional experiment, we assessed potential structural changes of bipolar cells using antibodies against PKCα for the IF staining [64]. The axon length of a bipolar cell is measured from under the soma to the axon terminal tip [33]. The information on primary antibodies is provided in Table A12. The immune reaction complexes were detected using secondary antibodies (goat anti-mouse IgG and goat anti-rabbit IgG) Alexa Fluor 488 (1:1000) diluted in goat serum. The sections were visualized using a confocal microscope (Zeiss LSM 980 with Airyscan 2, Zeiss, Oberkochen, Germany) following standard procedure. Three parts (central, mid-peripheral, and peripheral) of the retina of each section were collected according to previous literature [65]. The average thickness of the whole retina, as well as the length of bipolar cells, were quantified, as illustrated in Figure 5. Three to five cross-sectional slices were analyzed for each group.

### 4.8. Determination of Collagen Levels

Tissues including the cornea, lens, vitreous, retina, and sclera, of six-week-old rats of three different genotypes were carefully dissected and collected under a microscope after execution. The muscle and connective tissue attached to the cornea, lens, vitreous, retina, and sclera were separated with fine forceps as much as possible. The samples were homogenized in 6M hydrochloric acid and incubated at 95 °C for 20 h. After centrifugation (13,000× *g*, 10 min, 20 °C), hydroxyproline content in the supernatants was quantified using the Total Collagen Assay Kit (QZBtotcol1, QuickZyme, Biosciences, Rotterdam, The Netherlands) strictly following the standard protocol provided in the kit. For comparability between groups, the collagen content measured in each sample was normalized by the samples’ dry weight (collagen concentration/dry weight). The ocular tissue weight was provided in Table A3, which provides a standardized metric that facilitates accurate comparison of collagen content across different samples. Based on the consistency in processing methods for different groups, we adhere to the standard protocols of commercial assay kits and literature procedures [66] to quantify the total collagen content within tissues from rats of varying genotypes. Four animals were used in each group.

### 4.9. Determination of Collagen Ultrastructure in the Sclera

To investigate the morphology of collagen fibers, samples were obtained by cutting the sclera into a four-leaf clover shape, and approximately one mm^3^ was taken from each leaf surrounding the optic nerve within 1–3 min after execution and were fixed in fixative solution (G1102-100ML, Servicebio, Wuhan, China). The tissues were treated as previously described [67]. At least 200 collagen fibers were measured for diameter in each electron microscope image (Figure A2D). According to previous literature, the observation was performed using a transmission electron microscope (HITACHI, HT7800/HT7700) [68]. Three to five animals were used in each group.

### 4.10. Statistical Analysis

The allele number of the LoF *TIMP4* variants in individuals with high myopia and the population database were compared by Fisher’s exact test. The association between the LoF *TIMP4* variants and high myopia was calculated by odds ratio (OR), representing the effect size for Fisher’s exact test. The ocular parameters, amplitude of the ERG, thickness of the retina by HE, and height of bipolar cells were statistically analyzed using one-way ANOVA among three different genotypes of rats (*Timp4*^−/−^, *Timp4*^+/−^, and *Timp4*^+/+^). Before performing statistics analyses, the normal distribution (Shapiro-Wilk Statistic, *p* > 0.05) and homogenous variance (Levene test) were confirmed. When conducting post hoc analyses on the results of the one-way ANOVA, the Bonferroni correction method is employed to adjust for multiple comparisons. In instances of heteroscedasticity, Tamhane’s T2 correction is utilized to account for unequal variances. η^2^ represents the effect size for the one-way ANOVA. A paired T-test was used to compare the differences between the same rat before and after form deprivation. The detailed results of the analysis are provided in corresponding Appendix A. All the statistical results were calculated with SPSS 26.0 and plotted with GraphPad Prism 9.5.1. *p* < 0.05 is used as the basic statistical significance.

## 5. Conclusions

In summary, this study indicated that heterozygous LoF mutations in *TIMP4* are associated with high myopia, and *Timp4* deficiency disturbs rat ocular development through ECM remodeling by changing the biomechanical properties of the sclera and disturbing the structure and function of bipolar cells in the retina. Further research is warranted to confirm and expand upon these findings, deepening our understanding of the underlying myopia mechanism.

## Figures and Tables

**Figure 1 ijms-24-16928-f001:**
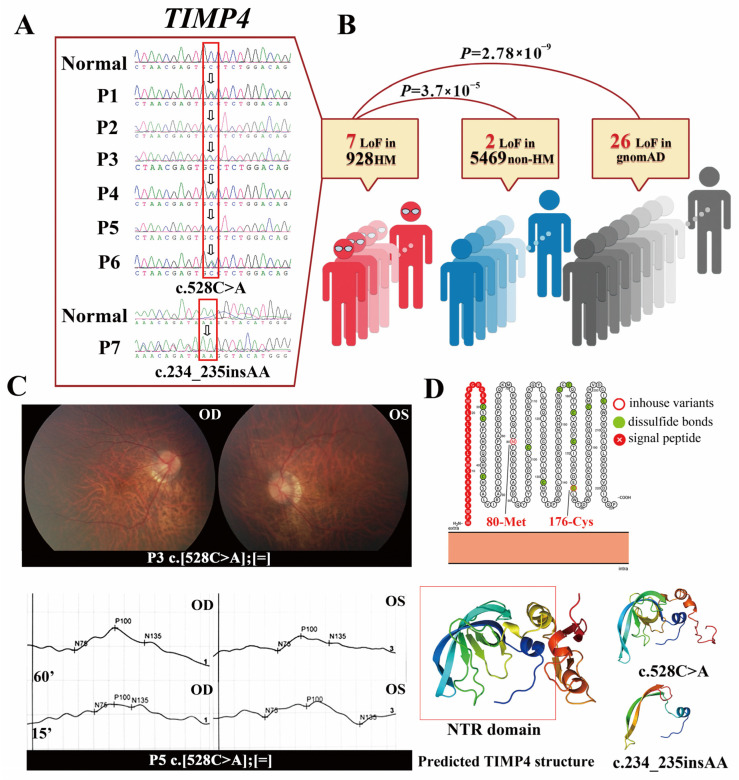
Variants and clinical information of individuals with high myopia in our cohort. Two potentially pathogenic variants in the *TIMP4* were revealed in this study. (**A**) c.528C>A was detected in 6 families, and c.234_235insAA was detected in one (indicated by white arrow in red box). The sequence of healthy controls and the corresponding sequences harboring the variants are shown. (**B**) Frequency of the *TIMP4* LoF variants in the HM population, non-HM population, and gnomAD database. The *TIMP4* LoF variants were enriched in the HM cohort. (**C**) Fundus photographs of P3 showed a large and pale optic disc and tigroid fundus of both eyes. The visual evoked potential (VEP) of P5 showed severe latency of 60 and 15 min-arc in OD, severe latency of 60 min-arc, and moderate latency of 15 min-arc in OS. (**D**) Predicted the TIMP4 structure and peptide chain structure of the corresponding variant. The position of c.528C>A is vital to disulfide bond formation, and c.234_235insAA results in the loss of the main structure, including the NTR domain, which is likely to cause NMD.

**Figure 2 ijms-24-16928-f002:**
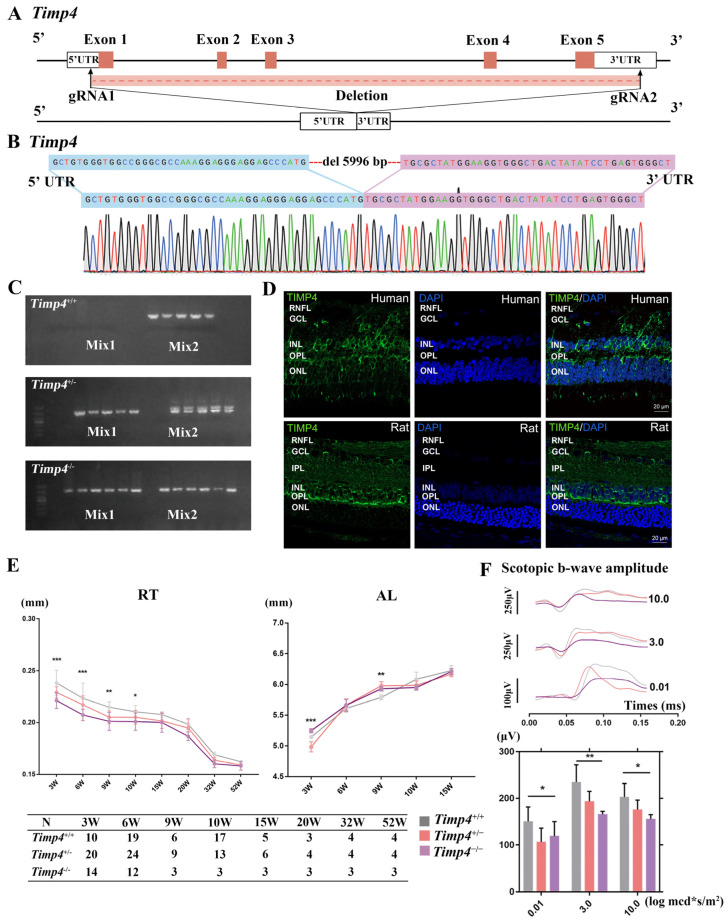
Knock-out of the *Timp4* in rats caused abnormal ocular morphogenesis and function. (**A**) Diagram illustrating the structure of knock-out of the *Timp4* in rat. (**B**) Sanger sequences confirmed the deletion of the whole *Timp4* gene in knock-out rats. (**C**) Interpretation of genotyping: *Timp4*^+/+^, zero bands in Mix1 and one in Mix2; *Timp4*^+/−^, one band in Mix1 and two in Mix2; *Timp4*^−/−^, one band in both Mix1 and Mix2. (**D**) The TIMP4 expression pattern in human retina and rat retina by immunofluorescence test. The scale bar is indicated by a short white line in the lower right corner of the figure (20 μm). (**E**) Retina thickness (RT) and axial length (AL) of the *Timp4*-deficient and wild-type rats decreased from 3W to 52W. The number of animals is provided in the table below. (**F**) ERG results of the *Timp4* deficient and wild-type rats at 3 w and 6 w. Compared with wide-type rats, a decrease in b-wave amplitude in 6-week-old *Timp4*-deficient rats is shown. Post hoc comparison: 0.01 log scotopic cd.s/m^2^: F = 4.876, η^2^ = 0.520, *p* = 0.037 < 0.05, *Timp4*^+/+^ vs. *Timp4*^+/−^ *p* = 0.041 < 0.05, *Timp4*^+/+^ vs. *Timp4*^−/−^ *p* = 0.637; 3.0 log scotopic cd.s/m^2^: F = 9.256, η^2^ = 0.673, *p* = 0.007 < 0.01, *Timp4*^+/+^ vs. *Timp4*^+/−^ *p* = 0.012, *Timp4*^+/+^ vs. *Timp4*^−/−^ *p* = 0.012; 10.0 log scotopic cd.s/m^2^: F = 5.347, η^2^ = 0.543, *p* = 0.029 < 0.05, *Timp4*^+/+^ vs. *Timp4*^+/−^ *p* = 0.069, *Timp4*^+/+^ vs. *Timp4*^−/−^ *p* = 0.040 < 0.05. The number of animals: *Timp4*^+/+^ = 4, *Timp4*^+/−^ = 5, *Timp4*^−/−^ = 4. N: number; W: week. A *p* value of less than 0.05, 0.01, and 0.001 was represented by an asterisk (*), two asterisks (**), three asterisks (***), respectively.

**Figure 3 ijms-24-16928-f003:**
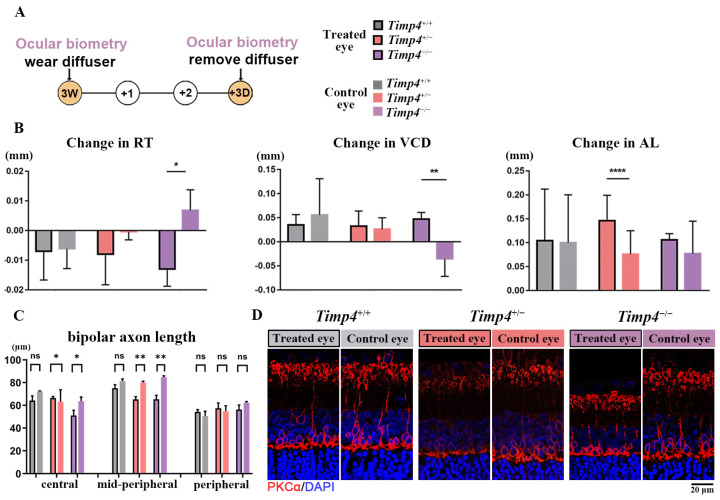
Ocular changes in the *Timp4* defect rats under form deprivation pressure. (**A**) Timeline of ocular biometry and form deprivation. (**B**) Change in RT, VCD, and AL of three-week-old rats before and after FD. The diffusers were removed after 72h. The FD eyes of the *Timp4*-deficient rats exhibited more severe retinal thinning, vitreous chamber deepening, and axial elongation than wild-type rats after FD induced. (**C**) The length of PKCα area in FD and non-FD eyes. (**D**) The IF staining of FD and non-FD eye in the mid-peripheral retina of three different genotypes. The red fluorescence represents PKCα and the blue fluorescence represents DAPI. The number of animals: *Timp4*^+/+^ = 6, *Timp4*^+/−^ = 5, *Timp4*^−/−^ = 4. A *p* value of less than 0.05, 0.01, and 0.0001 was represented by an asterisk (*), two asterisks (**), and four asterisks (****), respectively. A *p* value over 0.05 was represented by “ns”.

**Figure 4 ijms-24-16928-f004:**
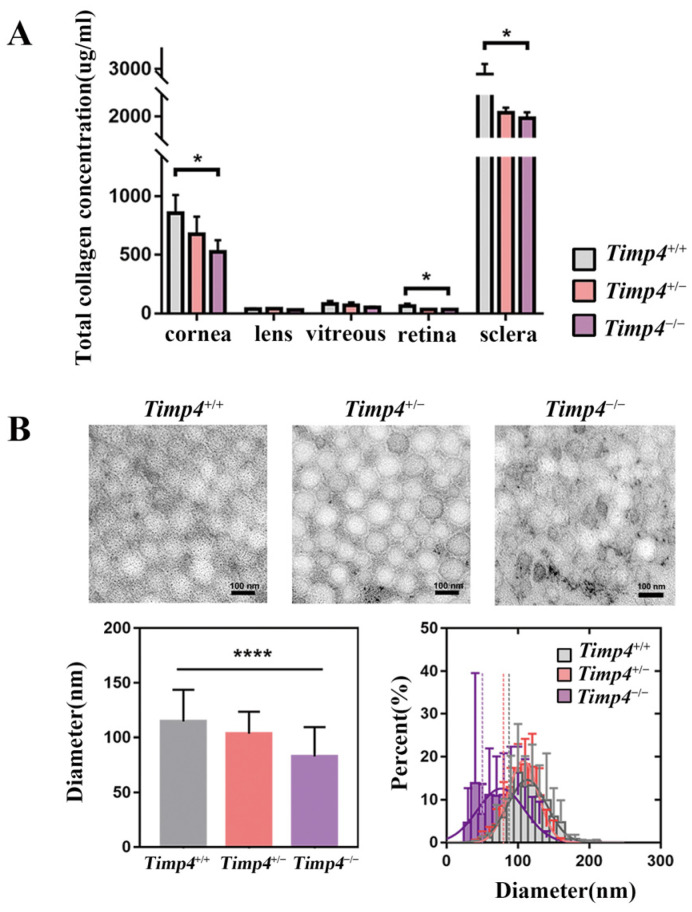
Morphological observation and concentration detection of collagen in different ocular tissues of 6-week-old rats. (**A**) Collagen content in eye tissues of 6-week-old rats, the collagen contents of the sclera, cornea, and retina in *Timp4*-knockdown rats were significantly lower than those in wild-type rats. The number of animals: *Timp4*^+/+^ = 5, *Timp4*^+/−^ = 4, *Timp4*^−/−^ = 3. (**B**) Diameter of scleral collagen in rats with different genotypes. The collagen fibers of the sclera in *Timp4*-deficient rats exhibited varying sizes, unsmooth edges, and larger gaps between adjacent collagen fibers compared with the wild-type. Sclera collagen fiber diameter in *Timp4*-gene editing rats was significantly smaller than in wild-type rats. One-way ANOVA, η^2^ = 0.1959, *p* < 0.0001, *Timp4*^+/+^ vs. *Timp4*^+/−^ *p* < 0.0001, *Timp4*^+/+^ vs. *Timp4*^−/−^ *p* < 0.0001. The number of animals: *Timp4*^+/+^ = 4, *Timp4*^+/−^ = 4, *Timp4*^−/−^ = 4. A *p* value of less than 0.05 and 0.0001 was represented by an asterisk (*), and four asterisks (****), respectively.

**Figure 5 ijms-24-16928-f005:**
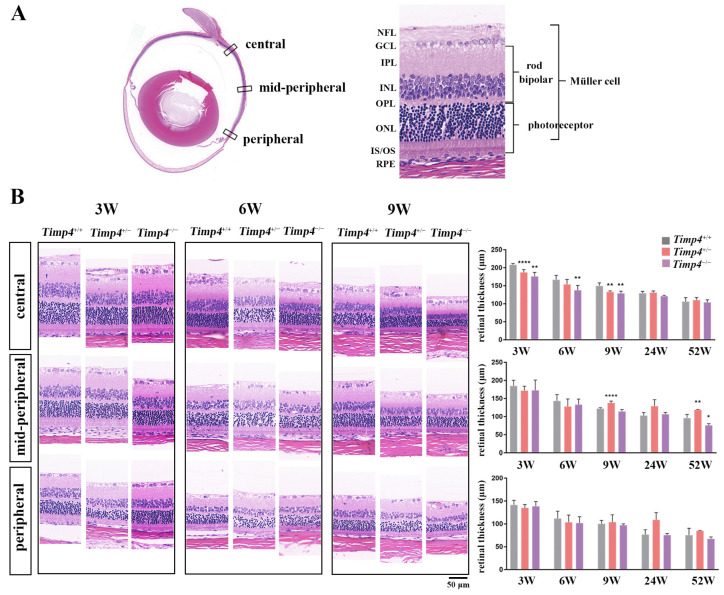
Morphology changes in the retina of *Timp4* defect rats. (**A**) Schematic diagram of slice observation and retinal layering. (**B**) The location of different cell types and the HE staining of the central, mid-peripheral, and peripheral retina at 3W, 6W, and 9W. Scale bar is indicated by a short black line in the lower right corner of the figure (50 μm). The central, mid-peripheral, and peripheral retinal thickness of the *Timp4*-deficient rats compared with wild-type rats at different ages. Detailed data is provided in Table A5. The number of animals: 3W, *Timp4*^+/+^ = 3, *Timp4*^+/−^ = 6, *Timp4*^−/−^ = 4; 6W, *Timp4*^+/+^ = 9, *Timp4*^+/−^ = 10, *Timp4*^−/−^ = 6; 9W, *Timp4*^+/+^ = 3, *Timp4*^+/−^ = 3, *Timp4*^−/−^ = 3. A *p* value of less than 0.05, 0.01, and 0.0001 was represented by an asterisk (*), two asterisks (**), and four asterisks (****), respectively.

**Figure 6 ijms-24-16928-f006:**
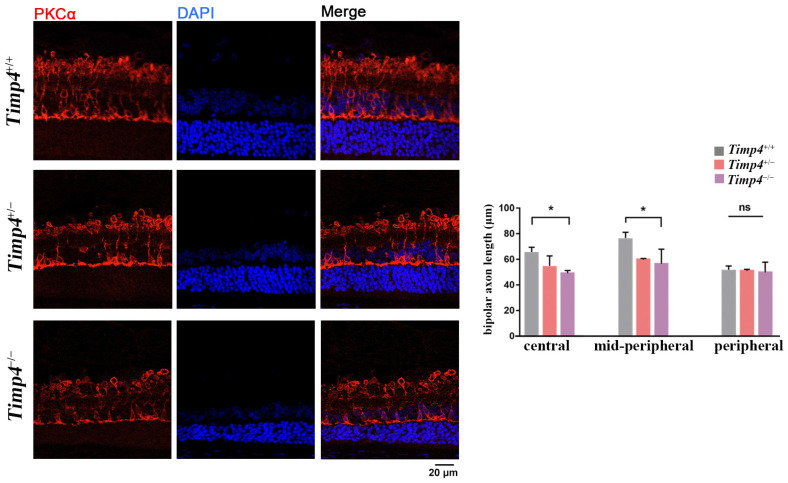
IF staining details in the central retina of six-week-old rats under the 40X confocal microscope and bipolar axon length of the central retina. Scale bar is indicated by a short black line in the lower right corner of the figure (20 μm). Red fluorescence: PKC-α (rabbit) & rabbit 568. The figure on the right shows that bipolar cells have shorter axon lengths in the *Timp4*-deficient rats than in wild-type rats in the central and mid-peripheral retina (One-way ANOVA, central: *Timp4*^−/−^ < *Timp4*^+/−^ < *Timp4*^+/+^, F = 9.032, η^2^ = 0.751, *p* = 0.0155 < 0.05, *Timp4^+/+^* vs. *Timp4^+/−^ p* = 0.0512, *Timp4^+/+^* vs. *Timp4^−/−^ p* = 0.011 < 0.05; mid-peripheral: *Timp4*^−/−^ < *Timp4*^+/−^< *Timp4*^+/+^, F = 5.702, η^2^ = 0.650, *p* = 0.041 < 0.05, *Timp4^+/+^* vs. *Timp4^+/−^ p* = 0.0696, *Timp4^+/+^* vs. *Timp4^−/−^ p* = 0.0343 < 0.05; peripheral: F = 0.06314, *p* = 0.9394). The number of animals: *Timp4*^+/+^ = 3, *Timp4*^+/−^ = 3, *Timp4*^−/−^ = 3. A *p* value of less than 0.05, was represented by an asterisk (*). A *p* value over 0.05 was represented by “ns”.

**Table 1 ijms-24-16928-t001:** Clinical features of high myopia probands carry the *TIMP4* LoF variants.

	Nucleotide Change		BCVA		RE		AL (mm)			
ID	(NM_003256.4)	Onset Age	OD	OS	OD	OS	OD	OS	Fundus	VEP
P1	c.528C>A	12	1.0	1.0	−8.00	−8.50	/	/	the bilateral temporal lens with cloudy anterior and posterior cortex, large optic disc with conus	/
P2	c.528C>A	2	0.5	1.0	−9.00	−6.00	/	/	large optic papilla, light temporal retina, and peripheral tigroid fundus, fovea reflex present	/
P3	c.528C>A	5	0.9-	0.9−	−6.25	−5.25	25.18	24.89	/	+
P4	c.528C>A	4	0.8-	0.8−	−10.25	−11.25	/	/	tigroid fundus, boundary line on the mid-peripheral retina	/
P5	c.528C>A	3	0.1	0.1	−11.25	−11.00	26.85	26.98	light nipple color, thin arteries, tigroid fundus, arc spots, poor retinal luster, peripheral retina degeneration, boundary line on the peripheral retina	+
P6	c.528C>A	12	1.0	1.0	−6.25	−7.25	26.37	26.83	conus and tigroid fundus	/
P7	c.234_235insAA	4	0.7	0.7	−10.75	−11.50	/	/	tigroid fundus with fovea reflex present	/

BCVA, best corrected visual acuity; RE, refraction error; AL, axial length; OD, right eye; OS, left eye; VEP, visual evoked potential; /, Not available; +, the incubation period is severely delayed, and the amplitude is normal.

## Data Availability

The data presented in this study are available on request from Wenhui Zhou, zhouwh23@mail2.sysu.edu.cn.

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
