# Peer review of "Defect of TIMP4 Is Associated with High Myopia and Participates in Rat Ocular Development in a Dose-Dependent Manner"

_ijms, 2023, doi:10.3390/ijms242316928_

Round 1
Reviewer 1 Report
Comments and Suggestions for Authors
Introduction:
The article titled "Defect of TIMP4 is associated with high myopia and participates in rat ocular development in a dose-dependent manner" explores the relationship between TIMP4 defects, high myopia, and their impact on rat ocular development. While the study has several strengths, there are notable areas for improvement.
Strengths:
Ethical Considerations and Approvals: The study begins on a strong ethical footing, emphasizing approval from the institutional review board and adherence to the principles of the Declaration of Helsinki. This demonstrates the commitment to ethical research practices.
Use of Whole Exome Sequencing (WES): The utilization of WES for candidate gene detection is commendable, particularly in genetic studies. Mentioning a dataset comprising a substantial number of probands enhances the credibility of the genetic analysis.
Standardized Ocular Measurements: Employing spectral-domain optical coherence tomography (SD-OCT) and electroretinography (ERG) for measuring ocular parameters and retinal function is a notable strength. These are established and scientifically valid techniques.
Appropriate Statistical Analysis: The study mentions the use of appropriate statistical methods, essential for ensuring the validity of research findings.
Methods:
While the methods section contains valuable information, there are areas where additional detail and clarity are required.
Areas of Improvement:
Incomplete Gene Knockdown Details: The section on Timp4 gene knockdown in rats lacks crucial information about the methodology of gene knockdown. Details on the delivery method of gene knockdown should be provided for clarity.
Form Deprivation Model: The introduction of the form deprivation model is mentioned, but the duration of form deprivation and how results were measured and analyzed are not provided. Detailed information about the form deprivation procedure and its impact on ocular development should be included.
Limited Information on Staining and Imaging: The section on HE staining and immunofluorescence lacks detailed information on staining and imaging procedures. Providing a more comprehensive description of these techniques would enhance reader understanding.
Insufficient Sample Details: The determination of collagen levels mentions the use of different ocular tissues but lacks details on sample sizes and how measurements were normalized. Including sample sizes and normalization methods would strengthen the validity of the results.
Data Presentation: The section should clearly state the significance threshold (e.g., p < 0.05) for the statistical analyses. Clarifying the level of significance applied would aid in understanding the statistical results.
Results:
The results section should present findings in a clear and organized manner, including statistical significance and effect sizes.
Discussion:
The discussion section should provide a thorough interpretation of the results in the context of the research question, identify limitations, and suggest avenues for future research.
In conclusion, while this study has potential, it requires substantial improvements in reporting methodological details and addressing other critical aspects such as participant information, gene knockdown methodology, and the form deprivation model. Strengthening these areas would enhance the credibility and reproducibility of the research.
Reviewer 2 Report
Comments and Suggestions for Authors
This paper evaluated the relationship between 33 MMP/TIMP genes to search for possible roles in early onset High Myopia. Based on this evaluation, this study identified that TIMP-4 plays a role in inhibiting the activity of MMPs. These studies demonstrated that TIMP4 Loss of Function mutations were associated with high myopia. In the knockout model, Timp4 deficiency effected the rate of ocular development through extracellular matrix (ECM) remodeling. Timp4 deficiency resulted in changes in the properties of the sclera, as well as the structure and function of bipolar cells in the retina. In this study, the Timp4-defect animal models also suggest that loss of Timp4 may play a role in cardiac function with aging, with Timp4-defect animals (rats) showing higher mortality due to myocardial infarction. This effect has already been reported in mice, as well as atherosclerosis in the abdominal aorta in Timp4-deficient rats. This paper could be strengthened by describing the role of Timp-4 and it’s binding partners, as well as what is already known in other animal models.
Revisions:
Page 6, Line 137 – change edinting to editing.
Page 6, Line 142 – change abstracted to extracted.
Page 6, Line 144 – change presented to present.
Page 9, Figure 3 – Figure 3D is too small. This image should be enlarges so the reader can see the images.
Page 13 – Figure 5 – this should be more than one figure of a sufficient size so that the reader can see the images. Figure 5B should probably be one image.
Page 14 – Line 313 – the cognition should be changed to the role of TIMP-4 in ocular tissue
Comments on the Quality of English Language
-
Reviewer 3 Report
Comments and Suggestions for Authors
of function (LoF) data of 6 early onset high myopia (Eo HM) patients and identified TIMP4 as an important player in determining changes in different ocular parameters. Those notion have been tested using a myopia model of rat. The ocular parameters such as vitreous chamber depth, retina, thickness and axial length in genetically modified in heterozygous and homozygous Timp4 KO rat. They concluded heterozygous loss of function mutations in Timp4 are associated with early onset high myopia, and the TIMP4 defect interferes with ocular development by influencing the morphology and function of the ocular tissue.
1. The method of Timp4 KO rat construction was under described. No information of the animal background, housing, sex have been provided.2. Show IF data as Neg, TIMP4 and DAPI. Then show merged images for both TIMP4/DAPI. The same comments apply for figures 2 and 3. Figure 3C and 3D data are very confusing. What is the length of PKCα area? No detailed information is available in the text. What was the implication of the length of PKCα area? What was actually probed for in Fig 3D? Are those PKCα/DAPI?
3. Table 2: (a) RT seems to be significantly different at age 3w and 6w and later the RT does not seem to be significantly thinner. Any explanations? (b) AL seems to be significantly different at age 3w and 9w and in the remaining time points it does not seem to be significantly different. Any explanations? (c) How were the p values calculated? Are those control vs homozygous KO or control vs heterozygous KO? The same comments on p value also apply for table 3.
4. The statistical data analysis throughout the manuscript appears very shallow. For example in figure 4B no data appear to be significantly different compared to the +/+ group. All error bars are highly overlapping. I recommend a statistical data reanalysis for the entire manuscript.
Round 2
Reviewer 3 Report
Comments and Suggestions for Authors
I am satisfied with the revised version of the manuscript along with the responses provided by the authors.